# MINING SHALLOW LAYER REPRESENTATIONS IN CLASS-INCREMENTAL LEARNING

## ABSTRACT

Class-Incremental Learning (CIL) aims to learn new knowledge without forgetting the old knowledge. One of the popular approaches is to obtain transferable representations, which would be general for learning incremental tasks without expanding the representations. Recently, many works focus on making the final representation more transferable across incremental tasks. However, researchers rarely focus on shallow layer representations and utilize their properties to facilitate CIL, although they are shown to be more transferable than the final representation. In this paper, we investigate the properties of the shallow layer representations and utilize them to improve the performance in class-incremental learning. Specifically, we show that shallow layer representations forget less than deeper layers. Furthermore, we find that shallow layer representations have more stable intra-class relations. Such intra-class relations reflect the task-agnostic information that the deeper layer representations lack. Therefore, we propose **In**tra-**c**lass **B**ackward **D**istillation (IncBD) to make the deeper layers learn from the intra-class relations of the shallow layer's representations, making the final representation more stable in terms of the intra-class relations. To compensate for the loss of class separability introduced by backward distillation, we also propose to train auxiliary classifiers for each layer's representation. Extensive experiments are performed to show that the intra-class relations are important for the transferability of the final representation and performance improvement in class-incremental learning.

## 1 INTRODUCTION

Deep models are good at capturing the necessary features of images for various tasks to form a compact representation. However, in real-world situations, new concepts and knowledge increase over time. It is necessary to allow deep models to adapt to new knowledge while keeping the previously learned knowledge. *Class-Incremental Learning* (CIL) is a scenario where new concepts incrementally emerge as new classes. When applying deep neural networks in incremental scenarios, the model usually forgets the previously learned knowledge, which is referred to as *catastrophic forgetting* (McCloskey & Cohen, 1989). Therefore, there are challenges to balance the model between *stability* (ability to resist changes) and *plasticity* (ability to adapt). To achieve this, researchers put their effort on regularization (Kirkpatrick et al., 2017; Shi et al., 2022), rehearsal memory management (Rebuffi et al., 2017; Liu et al., 2020), expandable block design (Yan et al., 2021; Douillard et al., 2022; Wang et al., 2022a; Zhou et al., 2023b), etc. Recently, there are works focusing on the final representation that the model has learned (Ramasesh et al., 2020; Zhu et al., 2021; Shi et al., 2022; Guo et al., 2022). These works regularize or design the model to approach some nice properties for the final representation, in order to obtain better transferability across incremental tasks. However, these works only focus on the final representation, ignoring the more transferable shallow representation in the same model (Yosinski et al., 2014; Ramasesh et al., 2020). To the best of our knowledge, the shallow layer representations are rarely utilized to facilitate the final representation in class-incremental learning.

In this paper, we investigate the properties of the shallow layer representations in CIL. Besides the better transferability, we show that the intra-class relations of the shallow layer representations are more stable than deeper layer representations during the incremental learning process. Specifically, we perform spectral analysis (Chen et al., 2019; Zhu et al., 2021) on each layer's representation for

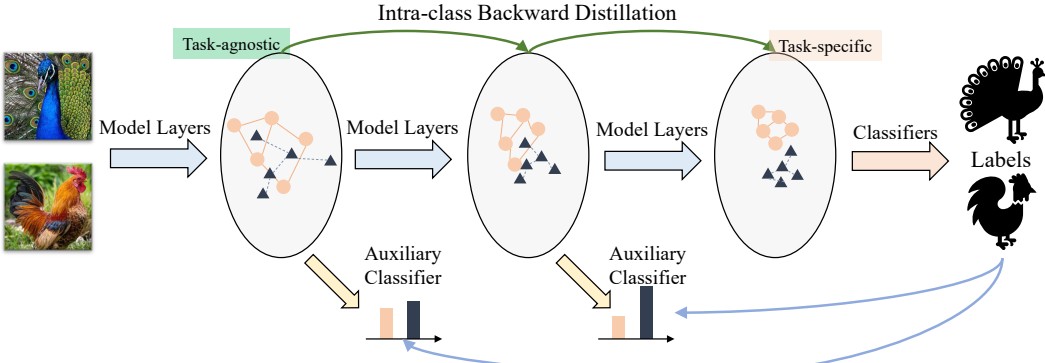

Figure 1: The general ideas of our work. The shallow layer representations have stable intra-class relational information, which contains the task-agnostic information the deeper layer representations lack. The proposed **In**tra-**c**lass **B**ackward **D**istillation (IncBD, green lines) makes deeper layer representations learn from shallow layer representations, carrying such task-agnostic information directly into the task-specific deeper layer representations, making the final representation transferable across incremental tasks. To compensate for the loss of class separability, we train auxiliary classifiers (blue lines) with shallow layer representations.

each class, finding that shallow layer representations have more transferable directions. Also, we use the intra-class relation distribution shift to investigate the relational information in each class for each layer's representation, showing that the relation distribution of shallow layer representations is more stable than deeper layer representations. Therefore, such intra-class relations reflect the task-agnostic information that the deeper layer representations lack; thus, the intra-class relations are crucial for the transferability of the representation and alleviating the forgetting during the incremental learning process.

To take advantage of the stable intra-class relations in shallow layer representations, we propose to make the deeper layers learn from the shallow layers via the intra-class relational distillation, which is what we call as **In**tra-**c**lass **B**ackward **D**istillation (IncBD). However, although the shallow layer representations show better transferability across tasks, they have less class separability, which impairs the classification performance. To compensate for the loss of class separability introduced by backward distillation, we propose to train auxiliary classifiers for each layer's representation. The general ideas are illustrated in figure 1. Extensive experiments are performed to show that the intra-class relations are important for the transferability of the final representation. The proposed IncBD improves the stability of the intra-class relations of deeper layer representations, also improving the transferability and performance in various CIL scenarios.

Our contributions are summarized as follows: 1) We study the layer representations for each class from two perspectives: the subspaces spanned by the representations of the whole class and the intra-class relations for each sample in the class, finding that the shallow layer representations are more stable in terms of both subspaces and intra-class relations. This analysis provides another perspective to understand catastrophic forgetting in CIL. 2) We propose IncBD and auxiliary classifiers to utilize the transferability of shallow layer representations, carrying the task-agnostic information directly into deeper layers. 3) Extensive experiments are performed to verify the effectiveness of our proposed methods.

## 2 RELATED WORKS

**Class-Incremental Learning (CIL)** is an incremental learning scenario where the model is learned task by task with a different set of classes. During inference, no task information about the samples is available. Many techniques and frameworks are proposed to alleviate catastrophic forgetting and improve the performance in CIL. They can be roughly categorized into several forms, such as model expansion, rehearsal memory, model distillation and regularization. Many of them can be seen as enhancing the learned representation for incremental tasks.

Model expansion comes from the idea of parameter isolation for each task. It expands the representation space as the task goes on. DER (Yan et al., 2021) trains a separate backbone for each

task, aggregating all of the representations for classification. DyTox (Douillard et al., 2022) learns a separate task token for each task. Rehearsal memory is used to store exemplars of previous tasks and replay at follow-up tasks. It makes the learned representation less forgetful by adjusting the input distribution towards the learned tasks. Many works focus on how to select exemplars (Rebuffi et al., 2017; Wu et al., 2019; Tiwari et al., 2022; Liu et al., 2020). Exemplars can also be obtained by generative models (Shin et al., 2017). Model distillation uses the model trained on previous tasks as a teacher and distillation losses to keep the previously learned knowledge in the representation. LwF (Li & Hoiem, 2017) proposes to use the response of the old model to guide the training of the new model's old tasks. PODNet (Douillard et al., 2020) uses the pooled intermediate feature maps of the ResNet to be the distillation target in training. Regularization methods come from various ideas, such as restricting the updates of important parameters (Kirkpatrick et al., 2017) to make the representation scatter uniformly (Shi et al., 2022).

Other perspectives to boost CIL are also considered. Zhu et al. (2021) proposes a dual augmentation framework to make the eigenvalues of the representation's covariance matrix larger. In the parameter space, Mirzadeh et al. (2020) studies the linear mode connectivity in CIL and proposes to enhance the linear mode connectivity between learned models. Lin et al. (2022) also considers the linear mode connectivity between learned models and proposes to combine two models learned in different ways to get better linear mode connectivity.

**Representation transferability in CIL**. There are several works (Ramasesh et al., 2020; Zhou et al., 2023b) find that shallow layer representations are much similar across tasks than deeper layer. Similar conclusions can also be found in transfer learning (Yosinski et al., 2014). Guo et al. (2022) uses contrastive learning to get holistic representations in online incremental scenarios.

However, these works only focus on the final representation, ignoring the more transferable shallow representation in the same model. The shallow layer representations are rarely utilized to facilitate the final representation in CIL to the best of our knowledge. In this paper, we further discover that the intra-class relations in shallow layer representations are more stable and important for cross-task transferability. The method proposed in our work is based on this observation and offers a new perspective on learning transferable representations in CIL.

## 3 PROBLEM FORMULATION AND EXPERIMENTAL SETUPS

### 3.1 PROBLEM FORMULATION

In class-incremental learning scenarios, we have multiple tasks to learn sequentially (Rebuffi et al., 2017). Let $D_t$ be the $t$th task. $(\boldsymbol{x}_i^{(t)}, y_i^{(t)}) \in D_t$ is a sample. $\boldsymbol{x}_i^{(t)}$ is the input, $y_i^{(t)}$ is the label. Let $\mathcal{C}_t = \bigcup_i \{y_i^{(t)}\}$ be the class set of task $t$. In CIL, $\forall t_1 \neq t_2, \mathcal{C}_{t_1} \cap \mathcal{C}_{t_2} = \emptyset$. In each task, we only train the model on $D_t$, but test on all the tasks the model has trained on, i.e., the presented tasks, without any of the other information, such as the task number that each sample comes from. For example, when the model is training on task $t_i$, the presented tasks are tasks $t$ ($t \leq t_i$), and they include all of the classes in $\bigcup_{t=0}^{t_i} \mathcal{C}_t$. The prevalent scenarios allow a fixed number of exemplars to be stored across tasks and trained together with follow-up tasks, which is called rehearsal memory replay. The final goal is to make the model get better performance on all of the presented tasks.

### 3.2 EXPERIMENTAL SETUPS

**Datasets**. Following most of the image classification benchmarks in CIL (Rebuffi et al., 2017; Wu et al., 2019), we use CIFAR100 (Krizhevsky, 2009) and ImageNet100 (Deng et al., 2009) to train the model. CIFAR100 has 50,000 training and 10,000 testing samples with 100 classes in total. Each sample is a tiny image in $32 \times 32$ pixels. ImageNet100 has 1,300 training samples and 50 test samples for each class.

**Data Split**. There are two common types of splits in CIL. The *small base* one equally divides all of the classes in a dataset (Rebuffi et al., 2017). The *large base* one uses half of the classes in a dataset as the base task (task 0), and equally divides the remaining classes (Hou et al., 2019; Yu et al., 2020). For a dataset with 100 classes in total, B10-10 means 10 classes in the base task and

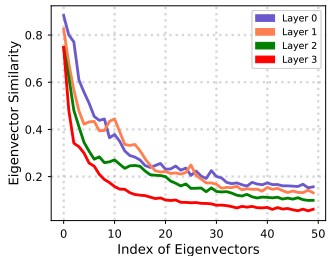 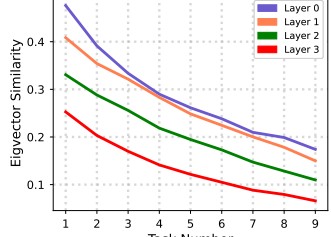 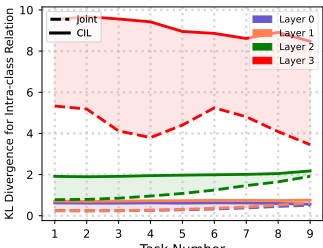

(a) Eigenvector similarity for each eigenvalue. (the largest 50 eigenvalues are shown)

(b) Eigenvector similarity for each task. (averaged across the largest 50 eigenvalues)

(c) KL divergence for intra-class relations for each task. Shadowed areas are the gap between Joint training and CIL.

Figure 2: Empirical analysis on layer representations. (a) Larger eigenvalues have more similar eigenvectors across the incremental training. Shallow layers suffer less representation shift. (b) The subspaces of shallow layer representations are more stable during the incremental training. (c) Shallow layer representations have much more stable intra-class relations across tasks. Joint learning scenario also shows less intra-class similarity distribution shift.

all of the following incremental tasks are also with 10 classes, B50-10 means 50 classes in the base task and 10 classes in the incremental tasks.

**Backbones and Baselines**. In this paper, we use four common backbones in vision tasks, ResNet32, ResNet18 (He et al., 2016), Convit (d'Ascoli et al., 2021), and PVTv2 (Wang et al., 2022b), which cover convolutional networks and vision transformers (ViT) (Dosovitskiy et al., 2021). Except for ResNet32, the parameter counts of the backbones are all around 11 million. The detailed descriptions and configurations of the backbones can be found in Appendix A.1. For all of the analysis in section 4 and 6.2, we use iCaRL (Rebuffi et al., 2017) with balanced classifier finetuning (Castro et al., 2018) for incremental tasks, and use 2000 maximum rehearsal memory capacity, and PVTv2 as the backbone, making it a comprehensive baseline covering most of the classical non-expanding incremental learning techniques. The representations at each layer are obtained by global average pooling. More detailed descriptions of the baselines and implementation details can be found in Appendix A.2 and A.3. Analysis results using other backbones and datasets can be found in Appendix B.2.

# 4 EMPIRICAL ANALYSIS ON LAYER REPRESENTATIONS

## 4.1 SPECTRAL ANALYSIS

In this section, we perform spectral analysis on each layer's representation in class-incremental scenarios. As mentioned in section 2, several works (Ramasesh et al., 2020; Zhou et al., 2023b) find that shallow layer representations are much more similar across tasks than deeper layers. Both of them use centered kernel alignment (CKA) (Kornblith et al., 2019) to measure the similarity between representations. However, we measure the representation space from another perspective, revealing the subspace spanned by the representations of each class. We then average the results across each class and present the results for each layer.

We use the spectral decomposition of the representation's covariance matrix to analyze the forgetting of each layer's representation for each class. To measure the similarity between representations, we follow Zhu et al. (2021) to compare the similarity of the subspaces spanned by the representations with corresponding angle (Chen et al., 2019) between the eigenvectors.

Specifically, suppose the representation of sample $\boldsymbol{x}$ for layer $l$ after training task $t$ is obtained by $f_t^{(l)}(\boldsymbol{x})$. We compute the covariance matrix for the representations of a group of samples and decompose it to different spectral angles:

$$\frac{1}{n}\sum_{i=1}^{n} f_t^{(l)}(\boldsymbol{x}_i)f_t^{(l)}(\boldsymbol{x}_i)^{\top} = \sum_{j=1}^{d} \boldsymbol{u}_j^{(l,t)}\lambda_j^{(l,t)}\boldsymbol{u}_j^{(l,t)\top}, \tag{1}$$

where $\lambda_j$ is the eigenvalue with index $j$ and $\boldsymbol{u}_j$ is its corresponding eigenvector. $d$ is the number of dimensions of the representation space. The indices of the eigenvalues are arranged from the largest

to the smallest. To compare the subspaces of two representation spaces, we use the corresponding angles of the eigenvectors:

$$\cos(\psi_j) = \frac{\langle \boldsymbol{u}_j^{(1)}, \boldsymbol{u}_j^{(2)} \rangle}{\|\boldsymbol{u}_j^{(1)}\|\|\boldsymbol{u}_j^{(2)}\|}. \tag{2}$$

The absolute value of this metric measures the similarity between two corresponding eigenvectors, representing the similarity of spectral directions of the representation spaces. With high eigenvector similarity between two representation spaces, the "shape" (i.e., covariance) is much similar. Therefore, we can use it to measure the representation shift between two tasks for each class. We compare the representation spaces of each class after each task with those after the task which the class is from (i.e., task 0 for classes 0-9 in B10-10). Results are summarized in figure 2a and figure 2b.

Figure 2a shows the eigenvector similarity for each eigenvalue averaged across the whole incremental training and each class. It shows that eigenvectors with larger eigenvalues are much similar throughout the incremental training. More importantly, the layer-wise results show that the deeper layer representations have less transferable directions compared to shallow layers, which means the deeper layer suffers more representation shift.

Figure 2b shows the eigenvector similarity for representation spaces after training each task, averaged across each class and each eigenvector with the largest 50 eigenvalues. It clearly shows that the subspaces of shallow layer representations are more stable and thus suffer less representation shift than deeper layers. Note that in this section, we show the summarized results of the PVTv2 backbone on CIFAR100. More detailed elaborations, plots, and results of other backbones and datasets can be found in Appendix B.

## 4.2 INTRA-CLASS RELATIONS

The spectral analysis investigates the class representations altogether, revealing the subspaces spanned by the whole class. To further study the properties of shallow layer representations, we perform a more in-depth analysis that considers the intra-class relations for a single sample. In this section, we investigate the intra-class relations for the representations of each sample. Formally, for each representation $\boldsymbol{r}_i^{(l)} = f^{(l)}(\boldsymbol{x}_i)$ of sample $\boldsymbol{x}_i$ from class $c$ at each layer, we compute the dot product with other representations in the same class, and take the softmax to get the intra-class similarity distribution for $\boldsymbol{r}_i^{(l)}$ within the class:

$$g_i^{(l)} = \text{softmax}\left(\frac{\boldsymbol{r}_i^{(l)} R_c^{(l)\top}}{\tau}\right), \tag{3}$$

where $\boldsymbol{r}_i^{(l)}$ is the $i$th column of matrix $R_c^{(l)}$, $\tau$ is the temperature hyperparameter of the softmax operation, we use the same $\tau$ for each layer. To compare the intra-class similarity distribution shift during the incremental training within the class for each layer, we compute the KL divergence between two similarity distributions, then average across the samples from the class:

$$\mathcal{D}_c^{(l,t_1 \to t_2)} = \frac{1}{N_c} \sum_{i=1}^{N_c} \text{KL}\left(g_i^{(l,t_1)} \| g_i^{(l,t_2)}\right). \tag{4}$$

To measure the intra-class similarity distribution shift, we compare the representations after training each task with those after the task which the class is from (i.e., $t_1 = 0$, $t_2 = t$ for classes 0-9 in B10-10). In addition to the incremental scenario, we also analyze the intra-class similarity distribution shift in the joint learning scenario, where we train each task with all of the samples from previous tasks to set a reference with upper bound performance. The results are summarized in figure 2c.

Figure 2c shows that the KL divergence for intra-class relation is much lower for shallow layer representations, meaning that shallow layer representations suffer less intra-class similarity distribution shift, and have much more stable intra-class relations across tasks. Also, the representations learned by joint learning always have less KL divergence for intra-class relations at each layer. Since the representation learned by joint learning is transferable to all of the tasks in the dataset, the results show that the preservation of intra-class relations is an important characteristic for transferable representations. Therefore, we can utilize this property to facilitate representation learning in CIL by making the deeper layer representations learn from shallow layers about their more stable intra-class relations to make the final representations more transferable.

## 5 METHOD

### 5.1 INTRA-CLASS BACKWARD DISTILLATION

Motivated by the results of empirical analysis in section 4, in this section, we introduce a simple yet effective technique called **In**tra-**c**lass **B**ackward **D**istillation (IncBD), making the deeper layers learn from the intra-class relations of the shallow layer's representations, in order to make the final representations more transferable. We can construct a similarity graph for each class and each layer's representation using equation 3. Then, the KL divergence loss can be applied to each layer and its prior layer:

$$\mathcal{D}_c^{(l)} = \frac{1}{N_c} \sum_{i=1}^{N_c} \mathrm{KL}\left(g_i^{(l-1)} \| g_i^{(l)}\right), \tag{5}$$

$$\mathcal{L}_{\mathrm{BD}} = \sum_{l=2}^{L} \mathcal{D}^{(l)}, \quad \mathcal{D}^{(l)} = \frac{1}{C} \sum_{c=1}^{C} \mathcal{D}_c^{(l)}, \tag{6}$$

where $C$ is the number of classes the model is training on, if there is a rehearsal memory buffer, $C$ equals the number of classes the model has seen so far; if there is no such buffer, $C$ equals the number of classes in the current task. Note that to avoid the inverse knowledge transfer, we stop the gradient propagation for $g_i^{(l-1)}$ in equation 5. Equation 5 requires averaging across all of the samples in the class, which is intractable in batch training. In practice, we only compute the average in a batch to approximate the actual average. IncBD loss directly makes the deeper layers learn from the prior layers about their intra-class relations, to alleviate the loss of task-agnostic intra-class relational information in the final representation.

Additionally, to make the approximation of the actual average more accurate, we extend the IncBD loss. We keep a queue to store the representations from the last several batches for each layer, so that the IncBD loss can use more samples in the class to construct the similarity distribution for each sample. However, since the model is constantly updating itself, the stored representations are not up-to-date. The mismatch would be severe when the queue gets longer. Therefore, there is a trade-off about the length of the queue. We investigate this in the hyperparameter analysis in section 6.3. The detailed elaborations of this queue can be found in Appendix C.

### 5.2 AUXILIARY CLASSIFIER

Although IncBD alleviates the loss of task-agnostic intra-class relational information in the final representation, it harms the class separability of the final representation in the current task, making the adaption to the current task more difficult. To compensate for the loss of class separability introduced by IncBD, we propose to train auxiliary classifiers for each layer's representation.

Specifically, for the sample $\boldsymbol{x}_i$ from class $y_i$ at layer $l$, its representation is $\boldsymbol{r}_i^{(l)} = f^{(l)}(\boldsymbol{x}_i)$. Note that we use global average pooling to get the representations at each layer. We train an auxiliary classifier $\phi^{(l)} : \mathbb{R}^{d_l} \to \mathbb{R}^C$ for each layer representations, using cross entropy loss:

$$\mathcal{L}_{\mathrm{aux}} = \sum_{l=1}^{L-1} \frac{1}{B} \sum_{i=1}^{B} \sum_{c=1}^{C} -\mathbb{I}(y_i = c) \log \mathcal{S}_c\left(\phi^{(l)}\left(\boldsymbol{r}_i\right)\right), \tag{7}$$

where $\mathcal{S}_c$ is a selection function selects the value at index $c$ of the input vector. For each task, the auxiliary classifiers are reset at the beginning of the task training. With the auxiliary classifiers, the class separability of the shallow layer representations can be improved, and more importantly, the side effects of IncBD can be alleviated. The effect of auxiliary classifiers and the trade-offs between IncBD are investigated in the ablation study in section 6.3.

### 5.3 OVERALL LOSS

To summarize our method, we use IncBD to make deeper layers learn from the prior layers about their intra-class relations, which is the backward distillation loss $\mathcal{L}_{\mathrm{BD}}$. To make the approximation of the actual average more accurate, we extend the IncBD with a representation queue for each layer.

Table 1: Performance Results on CIFAR100 with 2000 rehearsal samples if the method requires. Bold font represents our methods improve the baseline on the corresponding scenario.

| Scenario | | B10-10 | | B2-2 | | B50-10 | | B50-5 | |
|---|---|---|---|---|---|---|---|---|---|
| Backbone | Method | Last | Avg | Last | Avg | Last | Avg | Last | Avg |
| ResNet32 | LwF (Li & Hoiem, 2017) | 23.25 | 43.56 | - | - | 40.40 | 49.59 | 40.19 | 46.98 |
| | BiC (Wu et al., 2019) | 53.54 | 68.80 | 41.04 | 62.09 | - | 59.36 | - | 54.20 |
| | iCaRL (Rebuffi et al., 2017) | 50.74 | 65.27 | 36.62 | 56.08 | 47.20 | 57.12 | 44.80 | 52.66 |
| | LUCIR (Hou et al., 2019) | 43.39 | 58.66 | 37.09 | 56.86 | 54.30 | 63.17 | 50.30 | 60.14 |
| | PodNet (Douillard et al., 2020) | 41.05 | 58.03 | 32.99 | 51.19 | 54.60 | 64.83 | 53.00 | 63.19 |
| | DER (Yan et al., 2021) | 60.43 | 71.21 | - | - | 61.56 | 68.67 | 59.71 | 67.51 |
| | w/ IncBD & auxcls | **61.35** | **71.82** | - | - | **62.71** | **69.23** | **60.56** | **68.17** |
| | FOSTER (Wang et al., 2022a) | 61.20 | 73.14 | - | - | 64.23 | 71.12 | 60.46 | 68.81 |
| | w/ IncBD & auxcls | **62.24** | **74.54** | - | - | **65.20** | **72.25** | **60.72** | **69.52** |
| Convit | iCaRL w/ Convit | 60.59 | 74.39 | 56.54 | 71.46 | 64.53 | 71.25 | 61.55 | 70.13 |
| | w/ IncBD & auxcls | **64.54** | **76.16** | **57.12** | **72.84** | **65.16** | **72.69** | **62.01** | **70.37** |
| | DyTox+ (Douillard et al., 2022) | 62.06 | 75.54 | 57.09 | 74.35 | 66.75 | 73.36 | 64.70 | 71.30 |
| | w/ IncBD & auxcls | **66.41** | **76.79** | **57.62** | **74.42** | **68.03** | **74.78** | **65.22** | **72.83** |
| PVTv2 | iCaRL w/ PVTv2 | 66.71 | 78.28 | 59.79 | 74.31 | 65.06 | 74.62 | 60.54 | 71.63 |
| | w/ IncBD & auxcls | **68.34** | **80.16** | **60.84** | **75.81** | **66.72** | **75.43** | **63.44** | **73.26** |

(a) iCaRL w/ Convit CIFAR100 B10-10 (b) FOSTER ImageNet100 B50-10 (c) iCaRL w/ PVTv2 ImageNet100 B50-5

Figure 3: Test accuracies for each task during CIL training with different scenarios. More plots can be found in Appendix D.

To compensate for the loss of class separability introduced by IncBD, we propose to train auxiliary classifiers for each layer representation, which is the auxiliary classifier loss $\mathcal{L}_{aux}$. We add $\mathcal{L}_{BD}$ and $\mathcal{L}_{aux}$ with hyperparameter $\lambda_{BD}$ and $\lambda_{aux}$ to the loss function during each task training. The final loss would be: $\mathcal{L} = \mathcal{L}_{orig} + \lambda_{BD}\mathcal{L}_{BD} + \lambda_{aux}\mathcal{L}_{aux}$, where $\mathcal{L}_{orig}$ is the original loss of the CIL methods. It includes the final classification and distillation loss according to the specific CIL method. The effect of the hyperparameters is investigated in section 6.3.

## 6 EXPERIMENTS

### 6.1 PERFORMANCE EXPERIMENTS

To verify the effectiveness of our proposed methods, we test our methods on five different baselines, covering four different baseline methods: iCaRL Rebuffi et al. (2017), DER (Yan et al., 2021), FOSTER (Wang et al., 2022a) and Dytox+ (Douillard et al., 2022), and four different backbones: ResNet32 (for CIFAR100), ResNet18 (for ImageNet100) (He et al., 2016), Convit (d'Ascoli et al., 2021) and PVTv2 (Wang et al., 2022b). We report the test accuracy after the final task training (*Last*) and the average top-1 test accuracy across each task training (*Avg*), as shown in Table 1 for CIFAR100, Table 2 for ImageNet100. The baselines are introduced in section 3.2, and more detailed configurations and elaborations can be found in appendix A.1 and A.2. The experiments based on DER and FOSTER are implemented with the open-source code PyCIL (Zhou et al., 2023a). The experiments based on DyTox and ViT-based backbones are implemented with the open-source code of DyTox (Douillard et al., 2022). We will release our code upon acceptance.

As we can see from the results, IncBD with auxiliary classifiers (auxcls) consistently improves the CIL performance on three backbones in various scenarios. In some scenarios (e.g., DyTox+ in CIFAR100 B10-10, iCaRL w/ PVTv2 in ImageNet100 B50-5), our IncBD with auxiliary classifiers achieves around 4% performance improvement in the overall accuracy in the final task.

Table 2: Performance Results on ImageNet100 with 2000 rehearsal samples if the method requires. Bold font represents our methods improve the baseline on the corresponding scenario.

| Scenario | | B10-10 | | B50-10 | | B50-5 | |
|---|---|---|---|---|---|---|---|
| Backbone | Method | Last | Avg | Last | Avg | Last | Avg |
| ResNet18 | iCaRL (Rebuffi et al., 2017) | - | 67.06 | 53.60 | 65.44 | 49.10 | 59.88 |
| | LUCIR (Hou et al., 2019) | - | - | 60.00 | 70.84 | 57.10 | 68.32 |
| | PodNet (Douillard et al., 2020) | - | 63.96 | 67.60 | 76.96 | 65.00 | 73.70 |
| | CCIL-SD (Mittal et al., 2021) | - | - | - | 79.44 | - | 76.77 |
| | DER (Yan et al., 2021) | 65.48 | 75.06 | 73.08 | 79.71 | 72.46 | 78.61 |
| | w/ IncBD & auxcls | **66.06** | **75.31** | **73.86** | **79.92** | **73.08** | **79.10** |
| | FOSTER (Wang et al., 2022a) | 65.68 | 76.74 | 71.60 | 77.37 | 68.20 | 76.00 |
| | w/ IncBD & auxcls | **67.72** | **77.58** | **73.82** | **79.92** | **70.22** | **78.14** |
| Convit | iCaRL w/ Convit | 59.86 | 72.84 | 62.66 | 72.80 | 60.78 | 70.45 |
| | w/ IncBD & auxcls | **60.76** | **74.19** | **64.28** | **73.11** | **63.46** | **71.26** |
| | DyTox+ (Douillard et al., 2022) | 65.78 | 76.35 | 71.32 | 78.08 | 66.38 | 75.46 |
| | w/ IncBD & auxcls | **65.96** | **76.90** | **71.64** | **78.43** | **66.94** | **76.02** |
| PVTv2 | iCaRL w/ PVTv2 | 62.78 | 77.01 | 60.52 | 71.68 | 59.80 | 70.51 |
| | w/ IncBD & auxcls | **66.32** | **79.01** | **62.98** | **73.06** | **64.34** | **72.67** |

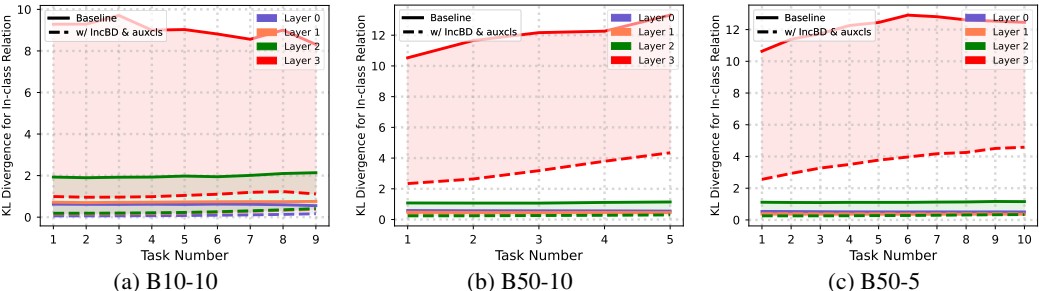

(a) B10-10  (b) B50-10  (c) B50-5

Figure 4: KL-divergence for intra-class relations for each task. Results for baselines are in solid lines, and results for our methods are in dashed lines. Shadowed areas are the gap between baselines and our methods.

Test accuracies for each task during CIL training with different scenarios are shown in figure 3. As we can see from the results, our methods have roughly the same test accuracy on the first task with baselines. However, on the follow-up tasks, our methods perform consistently better than baselines, showing that the model learns transferable representations from our methods. Test accuracies with more scenarios and the performance table with 500 rehearsal samples can be found in Appendix D.

## 6.2 INTRA-CLASS RELATIONS

To verify that the model gets more stable intra-class relations with our methods, we show the KL divergence for the intra-class relation as performed in section 4.2. We compare the similarity distribution for each sample in the same class after each task training with those after the first task training, to measure the similarity distribution shift for intra-class relations. We report the results averaged by each sample and class for each layer. The results are shown in figure 4 on CIFAR100 with PVTv2 backbone. We can clearly conclude that the intra-class relations are more stable with our methods, especially for the final representations.

## 6.3 ABLATIONS AND HYPERPARAMETER ANALYSIS

**Ablations**. In section 5, we proposed IncBD with auxiliary classifiers. To investigate their effects separately, we perform ablation experiments on CIFAR100 in B10-10 scenario with two backbones. The results are shown in table 3, and the accuracy by task is shown in figure 5. From the results, we conclude that in some of the CIL scenarios or backbones, IncBD or auxiliary classifiers alone do not lead to performance improvement (e.g., Convit backbone with 2000 rehearsal samples). Baselines with auxiliary classifiers get better accuracy in the first several tasks but worse accuracy in later tasks, showing that with auxiliary classifiers alone, the model would overfit to the first several tasks and forget more severely in follow-up tasks. However, we can get consistent performance improvement when IncBD works together with auxiliary classifiers, since they are complementary to each other.

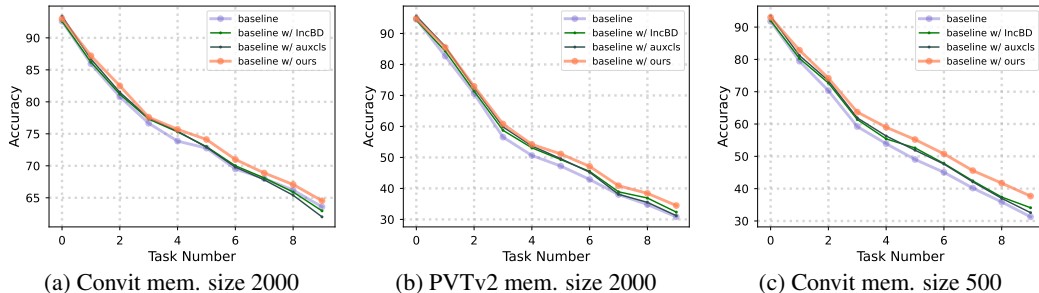

(a) Convit mem. size 2000     (b) PVTv2 mem. size 2000     (c) Convit mem. size 500

Figure 5: Ablation accuracy by task. All of the three experiments are on CIFAR100 in B10-10 scenario. Baselines with auxiliary classifiers get better accuracy but forget more severely in follow-up tasks. However, we can get consistent performance improvement when IncBD works together with auxiliary classifiers.

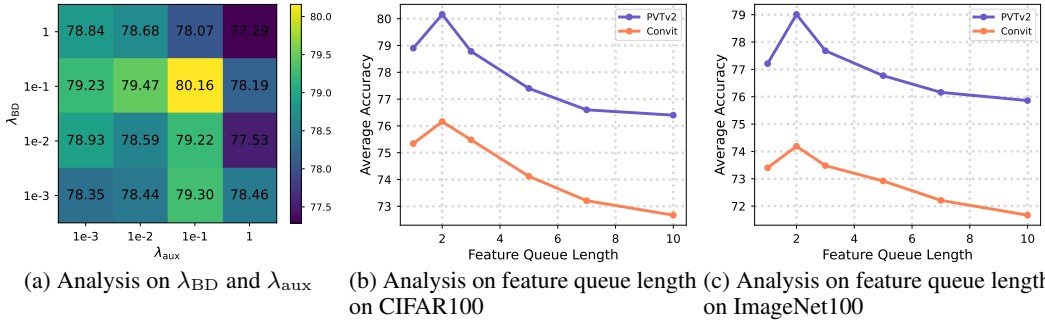

(a) Analysis on $\lambda_{BD}$ and $\lambda_{aux}$    (b) Analysis on feature queue length on CIFAR100    (c) Analysis on feature queue length on ImageNet100

Figure 6: Hyperparameter analysis. a) The best combination is achieve at both $\lambda_{BD}$ and $\lambda_{aux}$ are 0.1. b) and c) The best feature queue length is always achieved at the double size of a batch.

**Analysis on $\lambda_{BD}$ and $\lambda_{aux}$.** In the overall loss, we use $\lambda_{BD}$ and $\lambda_{aux}$ to combine IncBD and auxiliary classifiers into the training. To see how these hyperparameters affect the performance of CIL, we perform experiments with different combinations of a grid search. The results are shown in figure 6a. We report the average accuracy across all of the task training on CIFAR100 in B10-10 scenario. The best combination in this scenario is $\lambda_{BD} = 0.1$ and $\lambda_{aux} = 0.1$.

**Analysis on feature queue length**. For IncBD, we use a feature queue to make more samples available to construct the similarity distribution for each sample. To investigate how the length of the feature queue affects the performance of CIL, we perform experiments with different feature queue lengths. The results are shown in figure 6b and figure 6c. We study the feature length as multiples of the training batch size, and report the average accuracy across all of the task training in B10-10 scenario. The results show that the best feature queue length for both CIFAR100 and ImageNet100 with two backbones is always double the batch size.

Table 3: Ablation Study on CIFAR100 in B10-10.

| Memory Size | 2000 | | 500 | |
|---|---|---|---|---|
| | Last | Avg | Last | Avg |
| iCaRL w/ PVTv2 | 66.71 | 78.28 | 30.82 | 54.88 |
| w/ IncBD | 67.52 | 79.14 | 32.35 | 56.48 |
| w/ auxcls | 66.80 | 79.34 | 31.20 | 56.71 |
| w/ IncBD & auxcls | **68.34** | **80.16** | **34.48** | **58.02** |
| iCaRL w/ Convit | 63.59 | 74.99 | 31.27 | 55.61 |
| w/ IncBD | 62.96 | 75.26 | 34.09 | 57.56 |
| w/ auxcls | 62.02 | 75.24 | 32.56 | 55.98 |
| w/ IncBD & auxcls | **64.54** | **76.16** | **37.70** | **60.37** |

## 7 CONCLUSIONS

In this paper, we investigate the properties of transferable shallow layer representations in class-incremental learning from empirical perspectives. We perform spectral analysis and investigate the intra-class relations of layer representations. We find that the shallow layer suffers less representation shift in terms of the spanned subspaces. Also, we find that shallow layer representations have more stable intra-class relations than deeper layers. To utilize these properties of shallow layer representations, we propose IncBD to carry task-agnostic intra-class relations from shallow layers to task-specific deeper layers. Additionally, we use auxiliary classifiers to compensate for the loss of class separability. Extensive experiments are performed to verify the effectiveness of our methods and our methods get consistent performance improvement on various scenarios in CIL.

**Limitations**. The analysis focuses on non-expanding representations. The properties of concatenated representations are yet to be discovered.

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

# A    DETAILED EXPERIMENTAL SETUPS

## A.1    BACKBONE CONFIGURATIONS

The detailed backbone configurations are listed in table 4. Convit (d'Ascoli et al., 2021) is a ViT-based backbone featured with the gated positional self-attention (GPSA) layer, which introduces the convolutional inductive bias into ViT in a soft way. The PVTv2 backbone we use in this paper is PVTv2-B1-Li (Wang et al., 2022b), where "Li" means linear complexity attention layer, which reduces the number of parameters of PVTv2-B1 from 13.1M to 12.4M. PVTv2 is featured with pyramid-style vision transformer blocks, which extract the features of the image in different semantic levels. Extensive experiments are performed in this paper, covering both convolutional neural networks and ViT-based networks.

Table 4: Detailed backbone configurations.

| backbone | input size | | # of layers/stages | # of parameters | feature dimensions for each layer/stage |
| | CIFAR | ImageNet | | | |
| --- | --- | --- | --- | --- | --- |
| ResNet32 | 32 | - | 3 stages | 0.465M | 16, 32, 64 |
| ResNet18 | - | 224 | 3 stages | 11.2M | 128, 256, 512, 512 |
| Convit | 32 | 224 | 5 local layers, 1 non-local layer | 11.0M | 384, 384, 384, 384, 384, 384 |
| PVTv2-B1-Li | 128 | 224 | 4 stages | 12.4M | 64, 128, 320, 512 |

## A.2    CIL BASELINES

iCaRL (Rebuffi et al., 2017) uses herding (Welling, 2009) to select prioritized exemplars, and logit distillation between previously learned model and current training model for better representation learning. In this paper, we also finetune the model at the end of each task with class-balanced re-hearsal memory samples (Castro et al., 2018), making it a comprehensive baseline covering most of the classical non-expanding incremental learning techniques. DER (Yan et al., 2021) trains a separate backbone model for each incremental task, and uses expanded representations for prediction. FOSTER (Wang et al., 2022a) is based on feature boosting, which boosts the final representation for prediction. Dytox+ (Douillard et al., 2022) expands a task token for each task, with Mixup (Zhang et al., 2018) data augmentation to further improve the performance.

## A.3    IMPLEMENTATION DETAILS

For all of the experiments with convolutional neural networks (CNN), which is implemented based on the open-source code of PyCIL (Zhou et al., 2023a), the default experimental configurations for DER and FOSTER are untouched. We train the model for 200 epochs in the initial task and 170 epochs in the incremental tasks.

For all of the ViT-based experiments in this paper, which is implemented based on the open-source code of DyTox (Douillard et al., 2022), the base task and incremental tasks are trained for 500 epochs. In incremental tasks, we finetune the all of the classifiers with the balanced memory buffer for 20 epochs at the end of each task training. The learning rate of each task is set to 0.0005 with the cosine scheduler. For training, the standard DeiT (Touvron et al., 2021) augmentations are applied, also with Mixup (Zhang et al., 2018).

# B    MORE ELABORATIONS AND RESULTS ON EMPIRICAL ANALYSIS

## B.1    MORE ELABORATIONS ON EMPIRICAL ANALYSIS

As mentioned in section 4.1, we compare the representation spaces of each class after each task with those after the first task, to measure the representation shift during CIL. Formally, denote the $j$th eigenvector for the representations of class $c$ at layer $l$ after task $t$ as $\boldsymbol{u}_j^{(l,t)}$, and class $c$ is from task

$t_c$, so class $c$ first appears when training task $t_c$. We can get the eigenvector similarity by:

$$\mathcal{S}_j^{(c,l,t)} = |\cos \psi_j^{(c,l,t_c \to t)}| = \frac{\langle \boldsymbol{u}_j^{(c,l,t_c)}, \boldsymbol{u}_j^{(c,l,t)} \rangle}{\|\boldsymbol{u}_j^{(c,l,t_c)}\|\|\boldsymbol{u}_j^{(c,l,t)}\|}. \tag{8}$$

The results in figure 2a are averaged across each task and class, which is:

$$\mathcal{S}_j^{(l)} = \frac{1}{T}\sum_{t=1}^{T}\frac{1}{|\mathcal{C}_{0:t}|}\sum_{c\in\mathcal{C}_{0:t}}\mathcal{S}_j^{(c,l,t)}, \tag{9}$$

where $\mathcal{C}_{0:t}$ is the set of class indices from task 0 to $t$. The results in 2b are averaged across each class and the first 50 eigenvectors, which is:

$$\mathcal{S}^{(l,t)} = \frac{1}{50}\sum_{j=1}^{50}\frac{1}{|\mathcal{C}_{0:t}|}\sum_{c\in\mathcal{C}_{0:t}}\mathcal{S}_j^{(c,l,t)}. \tag{10}$$

Similarly, the results of KL-divergence for intra-class relation in figure 2c are firstly obtained by computing the KL-divergence between the averaged similarity distribution within class $c$ at layer $l$ after task $t$, which is:

$$\mathcal{D}_c^{(l,t)} = \frac{1}{N_c}\sum_{i=1}^{N_c}\mathrm{KL}\left(g_i^{(l,t_c)}\|g_i^{(l,t)}\right). \tag{11}$$

The results in figure 2c are averaged across each class, which is:

$$\mathcal{D}^{(l,t)} = \frac{1}{C_t}\sum_{c=1}^{C_t}\mathcal{D}_c^{(l,t)}. \tag{12}$$

### B.2 More Results on Empirical Analysis

In this section, we show the eigenvector similarity for each eigenvalue by task in figure 7. Using the notations defined in section B.1, figure 7 shows $\mathcal{S}_j^{(l,t)} = \frac{1}{|\mathcal{C}_t|}\sum_{c\in\mathcal{C}_t}\mathcal{S}_j^{(c,l,t)}$ for each task $t$. The figures show the eigenvector similarity for each eigenvector when we focus on the classes in only one task. From the results, we conclude that 1) The eigenvectors with small eigenvalues shift much quicker than larger ones, which is the layer-wise illustration of Zhu et al. (2021); 2) shallow layer representations shift much slower than deeper layers.

Figure 8 shows the eigenvector similarity for each task by task. Using the notations defined in section B.1, it shows $\mathcal{S}^{(l,t,t')} = \frac{1}{50}\sum_{j=1}^{50}\frac{1}{|\mathcal{C}_{t'}|}\sum_{c\in\mathcal{C}_{t'}}\mathcal{S}_j^{(c,l,t)}$, where $t'$ is the task we are focusing on (i.e., task number in the subcaption), $t$ are the tasks that task $t'$ appears to test on (i.e., task number on the x-axis). The figures show how the eigenvector similarity evolves when we focus on the classes in only one task. From the results, we can conclude that 1) the subspaces are less similar as the task goes on, reflecting catastrophic forgetting in CIL; 2) shallow layer representations have much more similar eigenvectors compared to deep layers, especially in the near subsequent incremental tasks.

We also perform empirical analysis on other backbones and datasets, which is shown in figure 9. From the results, we can conclude that the observations in section 4 are ubiquitous in different backbones. Thus, the conclusions are reliable.

## C More Elaborations on Feature Queue

In section 5, we extend IncBD with a queue storing representations of the last several batches for each layer. Formally, denote the queue for layer $l$ by $Q^{(l)} \in \mathbb{R}^{(nB,d)}$, where $n$ is the maximum queue length (by multiples of the batch size), $B$ is the batch size, $d$ is the number of dimensions of representation space. The queue $Q^{(l)}$ updates upon the forward pass of the model finishes for each batch, adding the representations of the newest batch to the end of the queue, and popping out the oldest batch if the maximum queue length exceeds, so that the queue contains the representations of the newest batch. The queue resets to empty at the beginning of each task. When computing the

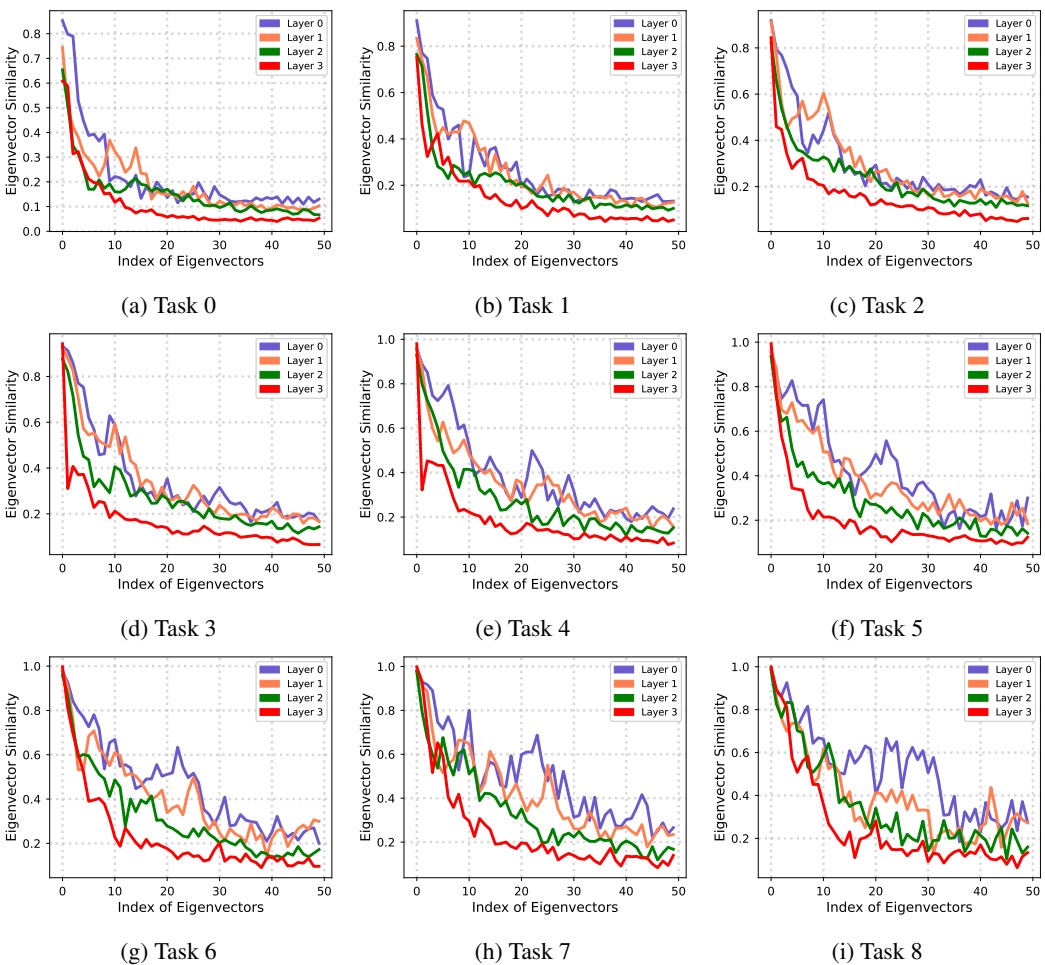

Figure 7: Eigenvector similarity for each eigenvalue by task. The curves for each task are averaged across its classes. The figures show the eigenvector similarity for each eigenvector when we focus on the classes in only one task.

Table 5: Performance Results on CIFAR100 with 500 rehearsal samples.

| Scenario | B10-10 | | B2-2 | | B50-10 | | B50-5 | |
|---|---|---|---|---|---|---|---|---|
| | Last | Avg | Last | Avg | Last | Avg | Last | Avg |
| iCaRL w/ PVTv2 | 30.82 | 54.88 | 13.55 | 42.50 | 24.23 | 47.16 | 17.59 | 40.07 |
| w/ IncBD & auxcls | **34.48** | **58.02** | **21.50** | **49.35** | **24.32** | **47.43** | **18.86** | **40.52** |
| iCaRL w/ Convit | 31.27 | 55.61 | 13.01 | 38.98 | 26.94 | 48.63 | 22.13 | 40.60 |
| w/ IncBD & auxcls | **37.70** | **60.37** | **14.02** | **43.51** | **28.88** | **49.04** | **24.60** | **42.02** |

intra-class similarity distribution for $\boldsymbol{r}_i^{(l)}$ in a batch, use the queue $Q^{(l)}$ instead of the representations in this batch $R^{(l)}$, which is:

$$g_i^{(l)} = \mathrm{softmax}\left(\frac{\boldsymbol{r}_i^{(l)} Q_c^{(l)\top}}{\tau}\right), \tag{13}$$

where $Q_c^{(l)}$ is a subset of $Q^{(l)}$ which only contains the representations of samples in class $c$ (i.e., the same class number with $\boldsymbol{r}_i^{(l)}$).

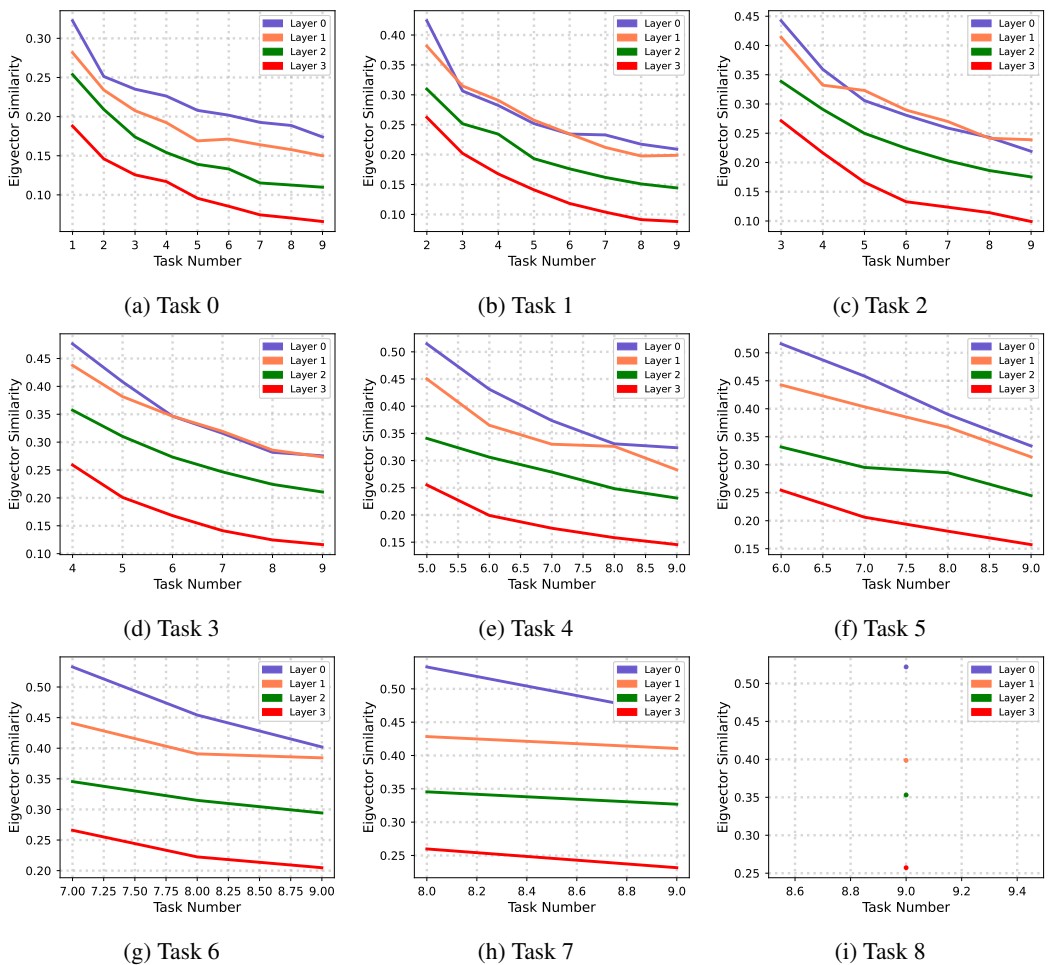

Figure 8: Eigenvector similarity for each task by task. The curves for each task are averaged across its classes. The figures show how the eigenvector similarity evolve when we focus on the classes in only one task.

## D  MORE RESULTS FOR PERFORMANCE EXPERIMENTS

We also test our method with 500 rehearsal samples, shown in table 5. In some scenarios, our methods achieve significant performance improvement (e.g., iCaRL w/ Convit in B10-10, iCaRL w/ PVTv2 in B2-2).

More test accuracies plots for each task during CIL training is shown in figure 10. We present the results with various scenarios, including different backbones, memory sizes, and datasets.

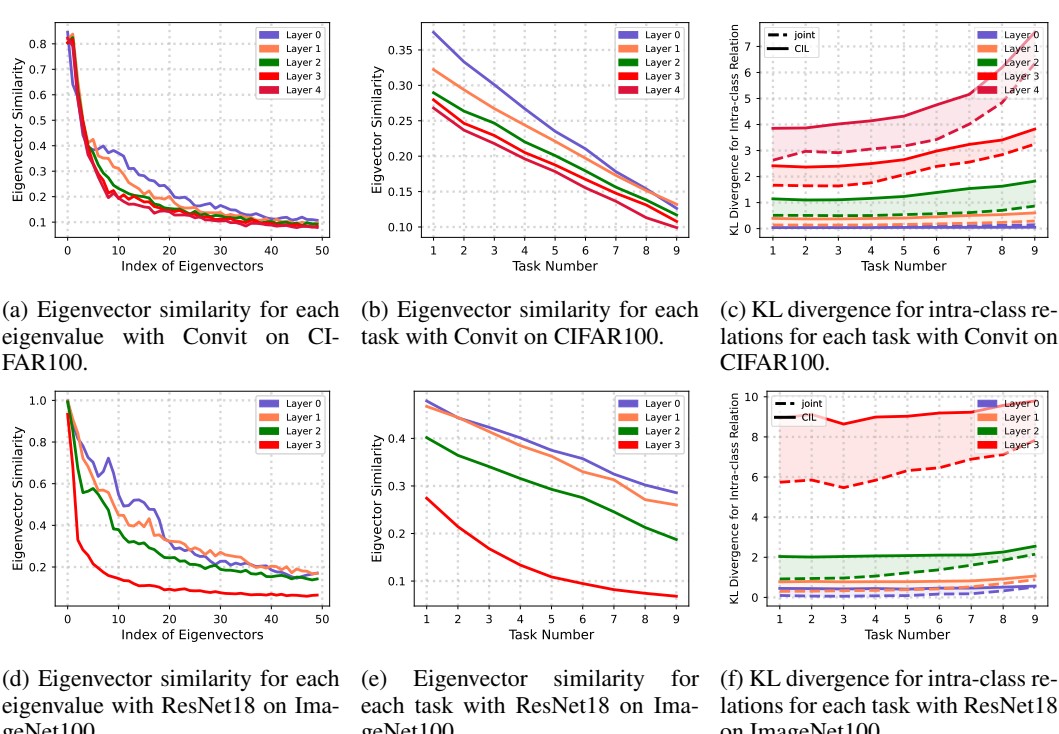

(a) Eigenvector similarity for each eigenvalue with Convit on CI-FAR100.

(b) Eigenvector similarity for each task with Convit on CIFAR100.

(c) KL divergence for intra-class relations for each task with Convit on CIFAR100.

(d) Eigenvector similarity for each eigenvalue with ResNet18 on ImageNet100.

(e) Eigenvector similarity for each task with ResNet18 on ImageNet100.

(f) KL divergence for intra-class relations for each task with ResNet18 on ImageNet100.

Figure 9: Empirical analysis on layer representations with different backbones and datasets.

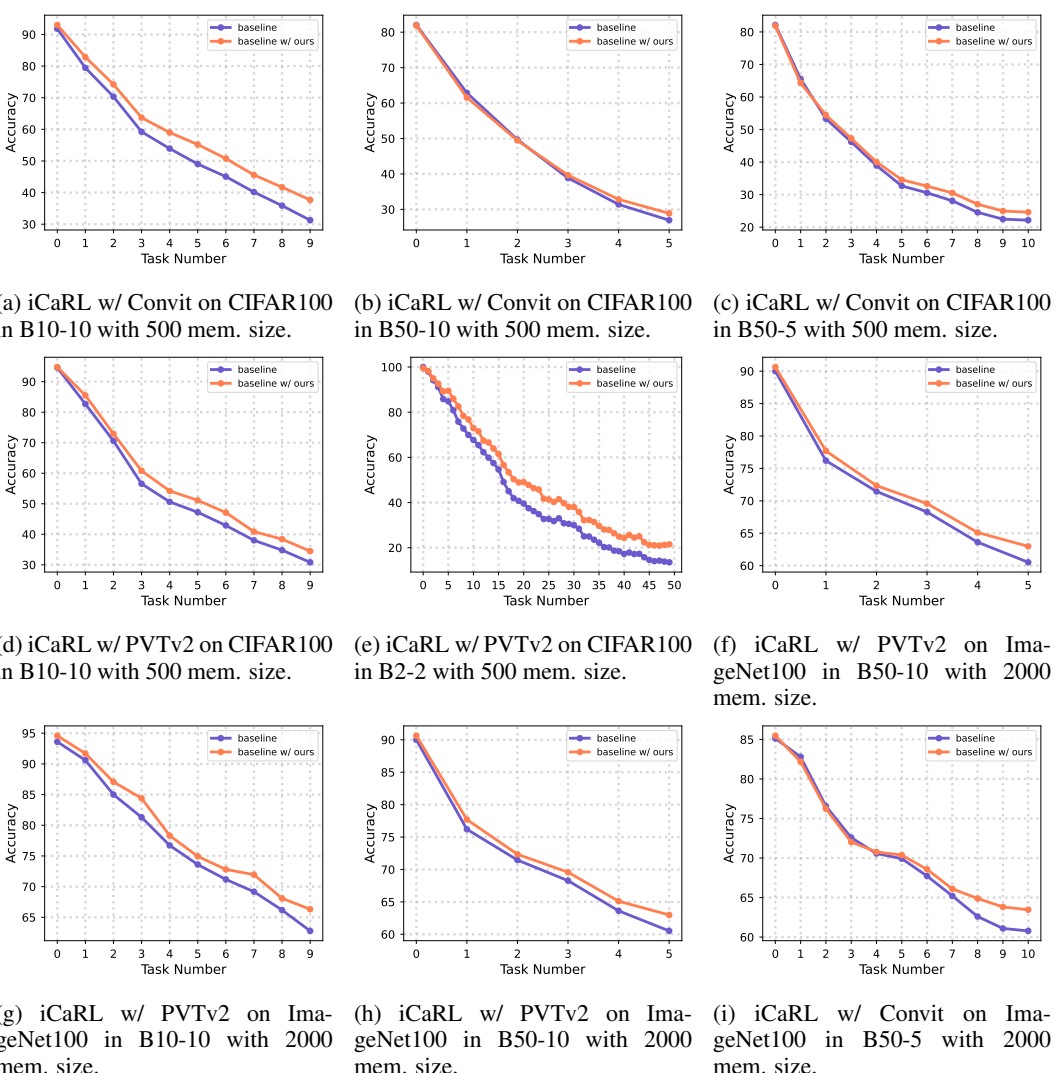

(a) iCaRL w/ Convit on CIFAR100 in B10-10 with 500 mem. size.

(b) iCaRL w/ Convit on CIFAR100 in B50-10 with 500 mem. size.

(c) iCaRL w/ Convit on CIFAR100 in B50-5 with 500 mem. size.

(d) iCaRL w/ PVTv2 on CIFAR100 in B10-10 with 500 mem. size.

(e) iCaRL w/ PVTv2 on CIFAR100 in B2-2 with 500 mem. size.

(f) iCaRL w/ PVTv2 on ImageNet100 in B50-10 with 2000 mem. size.

(g) iCaRL w/ PVTv2 on ImageNet100 in B10-10 with 2000 mem. size.

(h) iCaRL w/ PVTv2 on ImageNet100 in B50-10 with 2000 mem. size.

(i) iCaRL w/ Convit on ImageNet100 in B50-5 with 2000 mem. size.

Figure 10: Test accuracies results for each task during CIL training with various scenarios.

