# OpenReview forum: "Mining Shallow Layer Representations in Class-Incremental Learning"
_ICLR.cc/2024/Conference — ICLR 2024 Conference Withdrawn Submission_

### Official Review · Reviewer_RoqF · 2023-10-15

**Soundness:** 2 fair
**Presentation:** 2 fair
**Contribution:** 1 poor
**Rating:** 3
**Confidence:** 4

**Summary:**

This paper focuses on the task of class-incremental learning that aims to learn new knowledge without forgetting the old knowledge. And this paper indicates that many existing works pay more attention to making the final representation more transferable and ignore the shallow layer representations. To this end, this paper investigates the properties of the shallow layer representations and utilizes them to improve the performance in class-incremental learning. In the experiments, the proposed method is evaluated on multiple benchmarks.

**Strengths:**

Leveraging the spectral theory to make the corresponding analysis is interesting.

**Weaknesses:**

1. The motivation of this paper is somewhat unclear. In Introduction Section, the authors directly indicate that most existing methods only focus on the final representation, ignoring the more transferable shallow representation in the same model. I am not clear about the definition of the shallow representation. Besides, the authors should give more interpretations on why the shallow representation owns stronger transferable ability.

2. To the best of my knowledge, there exist some similar works that aim to leverage the shallow representation to enhance the generalization ability, e.g., self-distillation. The idea of this paper is not inspired, which lacks novelty. For example, the work [1] has observed shallow and deep layers have different characteristics in CIL. Based on these characteristics, the work [1] proposed an effective method to improve the performance of CIL. Thus, this paper employs a similar idea as the work [1], which does not propose an inspired idea.

[1] A Model or 603 Examples: Towards Memory-Efficient Class-Incremental Learning. ICLR, 2023.

3. The main contribution of this paper is to design a loss function, which is small. In the experiments, this paper should compare with state-of-the-art methods sufficiently, e.g., the works from CVPR 2023 and ICCV 2023. Besides, this paper aims to leverage the shallow representation to alleviate catastrophic forgetting. However, this paper does not provide any feature-level visualization analysis.

**Questions:**

This paper aims to leverage the shallow representation to alleviate catastrophic forgetting. However, this paper does not provide any feature-level visualization analysis.

---

### Official Review · Reviewer_MZ7v · 2023-10-30

**Soundness:** 3 good
**Presentation:** 3 good
**Contribution:** 2 fair
**Rating:** 3
**Confidence:** 5

**Summary:**

The paper proposes a method for class-incremental learning. The main insight is that layers closer to input generate representations which move less during incremental training. They perform an analysis to confirm this intuition. To exploit this, they introduce classifiers at multiple layers, and they propose Intra-class Backward Distillation (incDB) where the more stable layers are used to distill information to the 'deeper'(closer to output) layers. Results show that the proposed method can be combined with many methods and yields a performance gain.

**Strengths:**

- I like the main insight that layers closer to input are more transferable and that it should be possible to exploit this for CIL.
- the proposed analysis confirms that layers closer to the input change less and that they have more stable intra-class relations during incremental training.
- the method is general and can be applied to a wide range of ICL methods, and provides a nice performance gain for some settings.
- code would be released on acceptance.

**Weaknesses:**

- I understand that for each method the optimal hyperparameters have been selected ? (the 80.16 for PVTv2 (Table 1) was found to be optimal in Figure 6). If this is the case, that is not acceptable.

- Are the curves in Figure 5 made with the optimal hyperparameters for 'baseline w\ours'. If so, please put the optimal parameters for each of the methods. It would be nice to see all these curves in Figure 10 Appendix.

- Most of the used methods apply some form of distillation with the previous model. The proposed distillation based on 'lower layers' is shown to improve results, but you could also apply the intra-class distillation against the previous model (t-1). I am not convinced the proposed method is better than that.

- the paper should elaborate on early-exit networks which, like the proposed work, use auxiliary classifiers (however, they are used for a different purpose, namely fast inference. But papers show that having auxiliary classifiers yields better performance). See e.g.

[A] Shallow-Deep Networks: Understanding and Mitigating Network Overthinking Yigitcan Kaya, Sanghyun Hong, Tudor Dumitras

**Questions:**

- Please address the mentioned weaknesses, especially the one on the hyperparameters. For me, that is critical. If I wrongly understood this, I would raise my score. But the text seems to clearly state that the optimal parameters for each setting are chosen.

- also, more results on the ablation would be interesting. In the reference[A] they show that auxiliary classifiers in general improve results, however this is not seen in Figure 5.

minor:
- With respect to Eq (2) which compares eigenvectors with the same index, it seems very prone to noise. Also can you compare representations with different dimensions with each other ? (Maybe you should first map them to the same dimensions?) Since the curves in Figure 2a,b are based on different number of dimensions.

- Fig5 b ends with a performance of 34, what method is that in Table 1 ? PVTv2 seems to get 60?

---

### Official Review · Reviewer_7mgQ · 2023-10-31

**Soundness:** 2 fair
**Presentation:** 3 good
**Contribution:** 2 fair
**Rating:** 5
**Confidence:** 3

**Summary:**

The paper aims to improve the performance of class incremental learning by analyzing the representation property at shallow and deep layers of a neural network.
The paper first uses spectral analysis to demonstrate that “shallow layer representations are much more similar across tasks than deeper layers,” suggesting shallow layers are more transferable than deeper layers. The paper then analyzes intra-class relations based on examples from the same class and finds that the KL divergence of intra-class relations is lower in shallower layers and the joint-training setup. This suggests “the preservation of intra-class relations is an important characteristic for transferable representations.”
Based on the observations above, the paper proposes an “Intra-class Backward Distillation (IncBD)” approach to make the deeper layers learn from the intra-class properties of shallow layers.
More specifically, the proposed approach imposes a KL-divergence loss to make the deeper layers’ intra-class relation similar to that of lower layers. To compensate for the loss of discriminative properties as a result of the KL penalty, an auxiliary classifier is added to each layer.
Empirical results show that the proposed IncBD approach can consistently improve the performance of different class-incremental learning methods—DER, FOSTER, iCaRL, and DyTox+—in various scenarios.
Ablation studies demonstrate that both losses are required for successful class-incremental learning.

**Strengths:**

1. The analysis of shallow and deep layer representation dynamics in the context of class-incremental learning is interesting, especially when compared with the joint learning setup.

2. The paper presents strong empirical results, where IncBD consistently improves various class-incremental learning methods: DER, FOSTER, iCaRL, and DyTox+. This demonstrates that the proposed approach is robust and widely applicable.


3. The paper presents comprehensive ablation studies on loss weighting and feature queue length to analyze the impact of various algorithmic choices of the proposed approach.

**Weaknesses:**

My primary concern is that the theory regarding shallow/deep layers and the corresponding loss function proposal is not well-justified.

The transferability analysis on shallow and deep layers is not surprising—it is well-understood that the shallower layers learn more genetic/transferable representations while deeper layers learn more discriminative features. I'm unsure if I understand the main reasoning behind the proposed approach correctly: the paper appears to argue that shallower layers are more transferable; therefore, the deeper layers should learn from the shallower layers. The paper hints that lower intra-class divergence causes better joint task learning, but it is merely an observational finding, not a causal relationship. It remains unclear why the proposed inter-layer constraint and auxiliary classifier could improve learning. It is challenging to understand how the proposed approach makes the transferability and discriminability tradeoff to improve class incremental learning.

**Questions:**

1. Please summarize the main reasoning behind the proposed approach given the weakness section.
2. How to understand the two losses, KL and auxiliary classifier, in the context of the transferability and discriminability tradeoff? Why does the proposed approach produce a better tradeoff than vanilla deep neural network training?
3. Could you provide some intuition on the intra-class relation formulation? The proposed metric is based on dot-product and softmax. It is unclear why this could be a proper characterization for intra-class relations: dot-product, not other metric types, and softmax magnifies the differences in the metric.
4. How is $\lambda_{BD}$ and $\lambda_{aux}$ selected in each experiment?
5. As a potential ablation study, it would be interesting to know how the two losses impact the model across different layers by excluding some layers from contributing to the loss.

---

### Official Review · Reviewer_7Hxa · 2023-10-31

**Soundness:** 2 fair
**Presentation:** 3 good
**Contribution:** 2 fair
**Rating:** 3
**Confidence:** 5

**Summary:**

This paper addresses Class Incremental Learning (CIL) via exploring the potential of shallow layer representations. It introduces the Intra-class Backward Distillation (IncBD) technique, which allows deeper layers to learn from shallow layer intra-class relationships. This method aims to stabilize the final representation. To address potential separability issues, auxiliary classifiers are used for each layer. The experimental results demonstrate the effectiveness of this method.

**Strengths:**

The work is well motivated and is presented clearly.

**Weaknesses:**

(1)	Lack of Innovation
In CIL, existing methods emphasize feature distillation in the shallow layers of the network and have been proven to perform well, such as PODnet [Douillard A et al., 2020]. PODnet focuses on intra-layer knowledge distillation, while this paper distills knowledge between different layers. There is a lack of sufficient comparison with previous methods.

The analysis of features in different layers of the CIL model has already been presented in [Kim D et al., 2023]. From the perspectives of stability and plasticity, it is evident that the stability of the lower layers is higher. Thus the exploration of shallow layer representations is limited in novelty.

(2)	Experimental Results Do Not Support the Conclusion
The paper claims that the auxiliary classifiers can improve accuracy. Therefore, the performance improvement might be due to the increase in accuracy on the initial task, not because of reduced forgetting during CIL (see Figure 3-b). Why can't IncBD reduce forgetting without auxiliary classifiers (see Figure 5)? This does not support the insight of the paper.

(3)	Many Parameters Added, But with Only Marginal Improvement
Compared to the baseline method, this paper introduces two additional hyperparameters. At the same time, the improvement on ResNet, which is a primary setting in past works, is marginal. (0.21-0.49% on SOTA method DER)

[Douillard A et al., 2020] Podnet: Pooled outputs distillation for small-tasks incremental learning, ECCV2020.
[Kim D et al., 2023] On the Stability-Plasticity Dilemma of Class-Incremental Learning, CVPR 2023.

**Questions:**

Please refer to [Weaknesses]. Below are some suggestions regarding the experimental setup and results:
1) Lack of discussion on exemplar-free methods.
2) Lack of discussion on the number of samples preserved.
3) Some experimental results are hard to explain, such as in CIFAR100 B50-5, PODnet is higher than LUCIR by 2.7%, but in B50-10, PODnet is only 0.3% higher than LUCIR. This is different from the results of previous works like [Hu X et al., 2021, Chen X et al., 2023].

[Hu X et al., 2021] Distilling causal effect of data in class-incremental learning, CVPR 2021.
[Chen X et al., 2023] Dynamic Residual Classifier for Class Incremental Learning, CVPR 2023.